



# Brief Communication: Capturing scales of spatial heterogeneity of Antarctic sea ice algae communities

Alexander L. Forrest[1], Lars C. Lund-Hansen[2], Brian K. Sorrell[2], Isak Bowden-Floyd[3], Vanessa Lucieer[4], Remo Cossu[5], and Ian Hawes[6]

[1]Civil & Environmental Engineering, University of California – Davis, Davis, CA 95616, USA
[2]Department of Bioscience, Aquatic Biology and Arctic Research Centre, Aarhus University, DK-8000 Aarhus C., Denmark
[3]Australian Maritime College, University of Tasmania, Launceston, TAS 7250, Australia
[4]Institute for Marine and Antarctic Studies, University of Tasmania, Hobart, TAS 7053, Australia
[5]School of Civil Engineering, University of Queensland, St Lucia, QLD 4072, Australia
[6]Gateway Antarctica, University of Canterbury, Christchurch, Private Bag 4800, New Zealand

*Correspondence to*: Alexander L. Forrest (alforrest@ucdavis.edu)

**Abstract.** Identifying spatial heterogeneity of sea ice algae communities is critical to predicting ecosystem response under future climate scenarios. Using an autonomous robotic sampling platform beneath sea ice in McMurdo Sound, Antarctica, we measured irradiance in spectral bands expected to describe the spatial heterogeneity. Derived estimates of ice algae biomass identified patchiness at length scales varying from 50-70 m under first-year sea ice. These results demonstrate that a step-change in how these communities can be assessed and monitored. The developed methodologies could be subsequently refined to further categorize different ice algae communities and their associated productivity in both Arctic and Antarctic waters.

## 1 Introduction

Logistical challenges associated with the presence of ice cover makes estimations of algal biomass distribution and productivity difficult to quantify in large areas of the Southern Ocean. This is a significant limitation: sea ice covers up to 20 million $km^2$ of Antarctic waters (Holland et al., 2014) and conservative estimates attribute approximately 20% of primary productivity to algae that grow on or in sea ice (McMinn et al., 1999). To date, quantitative sampling of these ice algae has been largely resolved through the analysis of ice cores; a laborious and spatially constrained approach (Rysgaard et al., 2001). The lack of reliable broad-scale data on ice algae biomass distribution makes their contribution to regional food webs difficult to quantify, and thus potential consequences of climate-driven changes to sea ice dynamics on ecosystem functionality problematic to model.

Antarctic sea ice algae comprise a shade-adapted autotrophic assemblage, most frequently associated with the underside of sea ice during the austral spring and early summer (McMinn et al., 1999). The significance to marine ecosystems of a concentrated food resource prior to ice break-up, and its potential role in seeding pelagic production subsequent to ice break-up, are widely recognized (Arrigo & Thomas, 2004). To understand sea ice ecosystem function, it is critical to observe and measure these organisms at spatial scales matching the associated variability in their distribution and abundance. While the



existence of patchiness in sea ice algae is well known (Ambrose et al., 2005), techniques to quantify this patchiness over length scales of more than a few meters have been lacking. Research has recently begun to address this issue by developing algorithms to interpret the spectral modification of downwelling light passing through ice in terms of concentrations of photosynthetic pigments in the ice, a recognized proxy for biomass (Mundy et al., 2007; Ehn & Mundy, 2013).

Links between spectral attenuation and spectral changes through sea ice and algal biomass are well established (Arrigo et al., 1991), and algorithms to estimate biomass of sea ice algae from spectrally resolved irradiance have been developed in recent years in pace with remote sensing technologies. Most of these techniques have used empirically-derived Normalized Difference Indices (NDI); an approach widely used in satellite-based remote sensing of terrestrial vegetation (Townshend et al., 1985). NDI values have been developed for both Arctic and Antarctic regions and have been show to explain substantial

amounts of spatial variability of ice algae biomass (Mundy et al., 2007; Melbourne-Thomas et al., 2015).

## 2 Methods

*Site Selection* – From 5 – 8 November 2014, we further evolved this approach by using an Autonomous Underwater Vehicle (AUV) as a robotic survey platform to quantify the spectral properties of light transmittance through the ice at a 500 x 500m field test site approximately 1.5 km offshore of Cape Evans, Antarctica (77.63 S, 166.35 W; red filled circle – Fig. 1a). Cape

Evans was selected for this study as this relatively deep region (>220 m) of McMurdo Sound is well known for its uniform first year ice, dense algal growths and minimal snow cover (McMinn et al., 2000; Remy et al., 2008). At the time of sampling snow cover was absent, except for a few isolated patches a few meters across, around the region of Cape Evans (Fig. 1b). These patches were mainly very small (1s to 10s of meters) up to 5 cm thick surrounded by bare ice. To document the physical variability of the ice underside, the AUV was equipped with an upward looking Geoswath interferometric sonar

to characterize the topography at a resolution of 1cm vertical and 1m horizontal (Doble et al., 2009). Ice thickness was derived from collected sonar data and the recorded vehicle operating depth. Over the majority of the test site, ice thickness was fairly uniform, from 1.87-1.95 m with minor ridging and ice stalactite formation extending to ~2.4 m (Fig. 1c), characteristic of first year ice in the Southern Ocean (Dayton & Martin, 1971). Acoustic sonar measurements were supported using 14 stations where measurements of ice and snow thicknesses were collected using coring techniques along the three

AUV transect lines (CE1, CE2 and CE3; Fig. 1c).

*Sampling Design* – While AUVs have been previously used under-ice (Doble et al., 2009; Williams et al., 2014), this is the first such deployment to specifically measure properties of the transmitted irradiance through the ice cover. In contrast to Remotely Operated Vehicles (ROVs) that have been used for such a purpose before (e.g. Nicolaus & Katlein, 2013), AUVs operate autonomously and are a more stable platform capable of making measurements over longer distances with greater

spatial precision than ROVs. In this current work, we present a spatial analysis of transmitted spectral irradiance at different wave bands, collected at a sampling frequency of 25Hz using a self-recording Satlantic OCR507 multispectral radiometer



mounted on a *Gavia* AUV, across three separate 500 m return transects from a single deployment hole. At a preprogrammed speed of ~1.5 m s$^{-1}$, this corresponds to a single measurement every 6-8 cm, although the cone of acceptance of the sensor meant that, at the depth of travel (~6 m water depth), 75% of the irradiance sampled came from an under-ice footprint of approximately 4 m diameter. Data were therefore downsampled to 1 Hz to match the navigation sampling string of the vehicle and the spatial coverage of the irradiance sensor.

Measurements of spectral irradiance below the ice were collected at six 10 nm wavelength bands centered on 412, 470, 532, 565, 625 and 670 nm. Known absorption properties of sea ice and ice algae suggested that attenuation of lower wavelengths (412-532 nm) would result from snow, ice and ice-algae whereas at 565-670 nm attenuation would be primarily due to snow and ice (Hawes et al., 2012). To account for variable AUV depth and ice cover thickness, the spectral irradiance at each wavelength was calculated for the ice-water interface using a vertical exponential attenuation model. This combined wavelength-specific attenuation coefficients for transmitted irradiance (Kirk, 2011), measured on site, with AUV depth below the underside of the ice estimated from sonar measurements of overhead ice draft.

To develop a relationship between spectral under ice irradiance and areal chlorophyll-a (chla) biomass, we used elements of a concurrent survey being completed within the same study area as the AUV program. At 20 stations a TriOS Ramses ACC VIS cosine-corrected hyperspectral radiometer was deployed through a 250 mm diameter hole drilled through the sea ice. An arm was used to position the radiometer 1.2 m from the hole and approximately 10-15 cm below the ice. Spectra were taken towards the sun and at right angles to either side. Wavelengths from 320 to 950 nm are sampled by this instrument, at a resolution of 3.3 nm. We used raw TriOS under-ice irradiance in each of the wavebands measured by the Satlantic instrument by averaging the three values closest to 412, 470, 532, 565, 625 and 670 nm. NDIs were then derived as ($E_{\lambda 1}$ - $E_{\lambda 2}$)/($E_{\lambda 1}$ + $E_{\lambda 2}$) for each pair of wavebands, where $E_{\lambda 1}$ is irradiance at waveband 1.

An ice core was obtained from each position from where TriOS spectra were obtained using a 76 mm diameter SIPRE ice corer. The lower 50 mm of each, plus attached platelets, was cut off using a stainless steel hacksaw, and transferred to a shore laboratory in a dark, insulated box for processing within 2 h. Previous studies have shown that this length of the cut section of the core contains the majority of the chla biomass (>99% authors' unpublished data; see also Ryan et al., 2006, Carnat et al., 2014). Cores were imaged and characteristics of the ice core, specifically the presence or absence of embedded platelets, whether algae were primarily interstitial within the congelation ice or associated with platelets, and whether the core itself appeared to have been recovered with or without some degree of damage to the ice undersurface, was recorded.

In the laboratory, ice cores were thawed in the dark after adding two volumes of melted seawater to minimise the effects of osmotic shock. Immediately after thawing, the final volume was recorded, cell aggregations broken up by repeatedly passing them through the nozzle of a 60 ml syringe, and aliquots were filtered onto Whatman GFF filters for analysis of chla. Filters were stored individually wrapped and frozen in liquid nitrogen for up to four weeks during return to Denmark, whereafter storage was at -80$^{\circ}$C. For chla analysis, filters were extracted into 95% acetone and pigment concentration determined using



a Shimadzu UV2401 twin-beam spectrophotometer at 665 and 750 nm. Subsamples of ice algal suspension were preserved in Lugol's iodine for determination of dominant algae.

*NDI Development* – After combining the TriOS spectroradiometer-derived wavebands pairwise to derive each possible NDI, we used a Pearson correlation with ice core-derived chla, after log transformation, to determine which best predicted the chla. To minimise the effects of small scale heterogeneity that was apparent in the collected images of the ice underside, we used an averaged NDI for the three values measured at each of the 20 stations, these being collected from within 1.7 - 2.4 m of each other. Likewise, we calculated averaged chla for all cores from each station that were not obviously damaged during retrieval. Damaged cores were typically outliers in the NDI-chla relationship. Assuming that damage to ice cores would lose rather than gain biomass, we selected the replicates with the highest chla concentrations when there was suggestion of a damaged ice core and high inter-replicate variability.

We then calculated the NDI values for each point along the AUV transects from the on-board Satlantic instrument for estimating chla. As the wavebands of the TriOS and Satlantic were not identical, rather than directly using the TriOS-derived NDI:chla relationship with the Satlantic-derived NDI, we elected to tune the calibration of AUV transects by scaling chla estimates to reproduce the overall mean and range of observed concentrations from the 60-sample ice core dataset, after excluding outliers from potentially damaged cores.

## 3 Results

Chlorophyll-a concentrations in the training dataset were log-normally distributed with a geometric mean of 26.7 mg m$^{-2}$, and 95% confidence interval of 11.4 – 62.5 mg m$^{-2}$, similar to previous values and ranges from this location (McMinn et al., 2000). In all samples, the algae were primarily diatoms, with *Fragilariopsis obliquecostata*, a common diatom in Antarctic summer sea ice (McMinn et al., 2000), dominating 80% of the ice cores. *Navicula directa* and *Nitzchia frigida* were subdominants. Diatoms occurred both interstitially in the platelet ice matrix (Fig 2), and of the cores analysed, 29 of 60 were characterized as primarily algae interstitial within congelation ice and 31 as primarily associated with embedded platelets. Fig. 2c shows the complexity of the under-ice habitat and one of the large ice blocks (~1 x 1 x 2 m) removed to allow access for the AUV (Fig. 2d).

Using the paired, TriOS-derived wavebands and chla data, we determined that an NDI of [$E_d$(470)-$E_d$(565)]/[$E_d$(470)+$E_d$(565)] explained the highest proportion of the variability in algae biomass (N = 19, r = 0.63, p = 0026; Fig. 3). These wavebands correspond to areas of the spectrum that penetrate cold ice relatively well (Perovich, 1990), while 470 nm is close to the peak of ice algal absorption whereas 565 nm is close to the minimum algal absorption (Hawes et al., 2012; Melbourne-Thomas et al., 2015), and thus this NDI can be expected to be sensitive to biomass. At least one other investigation has found ratios constructed from similar wavelengths to be effective predictors of chla concentrations (Fritsen et al., 2011). Other studies that have linked optical properties with biomass (Mundy et al., 2007; Melbourne-Thomas et al.,



2015; Gutt, 2001), have often found close-spaced wavelengths within the part of the spectrum most attenuated by diatom pigments to produce NDI's that better describe variability in chla than our relationship. However, our use of the Satlantic instrument, while fully allowing autonomous operation, limited the wavelengths that were available. We consider that the most likely explanation for the relatively poor prediction power of our NDI compared to other published relationships is in the variability of chla concentrations at the decimeter scale of the ice cores (evident in Fig. 2d), in comparison to the much larger spatial scale of AUV optical footprints (~ 4 m diameter). Analysis of this issue is ongoing and will be published elsewhere. Snow and multi-year ice were largely absent from the study area, but it is possible that other substances, including windblown dust and mineral precipitates, could have contributed to spatial variability in optical properties. However, spectral scans of both under-ice irradiance and particulate material from the ice collected onto filters, was dominated by signatures of ice algal pigments (data to be presented elsewhere). For the purpose of evaluating the value of AUV observations to map spatial variability in sea ice algae, we consider it possible to assume that spatial variance of the spectral qualities of transmitted irradiance and the NDI to be primarily driven by variability of chla biomass.

Examples of derived spatial variability in chla concentrations are presented from repeat runs of CE1 at 23:45 04 Nov 2014 and 09:00 05 Nov 2014 (local times; Fig. 4a) and agree closely at the spatial sampling scales with a degree of variability resulting from differing light levels. Results are also presented of chla values on CE2, demonstrating the lack of relationship between minor variations in the first year ice topography (shown in grey) and the observed variations in ice algae biomass (Fig. 4b). While there is not enough evidence to be conclusive, there are reduced concentrations at the ice stalactite slightly increased concentrations around the less sharp ice perturbation into the water column at around 360 m. Measurements along parallel tracks, approximately 20 m apart, show similar amplitudes and length scales of variation (e.g. CE3; Fig. 4c) that would be expected to be poorly sampled using surface-based techniques.

## 4 Discussion

Both measured spectral irradiance below the ice and estimated chla showed varying amplitudes and length scales of patchiness. For each transect, chla estimates were interpolated to constant spacing between samples to allow a spectral analysis of the data using a standard Fast Fourier Transform (FFT) similar to time series analysis. A spectral analysis of the length scale of chla variation for the independent transects revealed the dominant scale to be 56.5 ± 17.5m (n = 6; σ represents one standard deviation). This compares favorably with similar estimates based on correlograms, constructed for each transect by plotting autocorrelations ($r_h$) against offset distance ($h$), from 1 to 150 space intervals. Dominant values of $h$, where $r_h$ was locally maximal, indicated length scales of patchiness varying from 21 to 66 m. Reproducibility of the length scale of chla variation between transects and resolution along transect to this degree has previously been unavailable and contributes to an instrumental understanding of the spatial heterogeneity of the ice algal biomass. They also indicate that coring-based sampling strategies designed to capture biomass spatial distribution need to be undertaken over similar spatial





scales, if they are to capture ice algae spatial heterogeneity sufficient to make robust contributions to understanding the contribution of ice algae to ecosystem function.

Variability in ice algal biomass has been recognized at a range of length scales (Melbourne-Thomas et al., 2015; Ambrose et al., 2005). Our data are, to our knowledge, unique in that they provide continuous lines of coverage from which the length scales of variance can be estimated. The mechanism underlying the variability of ice algae biomass, and the apparently rather consistent length scales of autocorrelation are not immediately apparent. Extracting the overhead ice draft measurements from CE2 at 12:00 05 Nov 2015 (Fig. 4b), we found no significant correlation between ice or snow thickness and NDI-derived chla estimates, and neither was a consistent relationship observed between chla and ice thickness or snow cover in the calibration core dataset. Given the virtual absence of snow from the ice cover, this is not surprising but it is possible that the pattern that we saw at the time of sampling was a legacy of snow conditions prior to the survey. Other variables with potential impact on biomass accrual over such length scales include those that affect colonization, growth and loss processes. Stochastic processes that determine the initial colonization of the ice surface, followed by time for expansion since patch inception, may in part be responsible for the consistent length scale. Alternately, the spatial heterogeneity may be an indirect result of physical processes within the under-ice boundary layer that create patterns in the suitability of the under-ice surface for algal growth, for example the extent of embedded platelet crystals that enhance the surface area for algal growth and provide an open structure for the delivery of nutrients.

While the developed relationship for NDI is important for this study, the lack of a more robust direct calibration of the NDI to chla, using the same optical instruments as on board the AUV, limits the interpretation of spatial variance in optical properties in terms of chla to the conditions associated with our study site. However, in ice of near-uniform thickness and largely free of snow, with imagery to support the perception of biomass patchiness we consider our assumption of a dominant effect of chla consistent with other published observations (e.g. Melbourne-Thomas et al., 2015; Mundy et al., 2007) and a useful demonstration of the utility of AUV-based observations in addressing spatial variability in sea ice algae. Demonstration of similar scales of patchiness within ice algal populations for other sea ice regimes where, for example optical properties of ice may be different, a greater proportion of ice algal biomass may be associated with infiltration ice, or snow cover may be more extensive (Gutt, 2001), will require further validation before our approach to documenting spatial variability using autonomous underwater vehicles and an NDI is universally applicable. These differences in ice properties make universal algorithms for converting optical properties of ice to algal biomass elusive (Melbourne-Thomas et al., 2015), though with the development and testing of area-specific algorithms the acquisition of optical data using AUV techniques should be as equally applicable as other optical methods.



## 5 Conclusions

This work demonstrates pioneering work using a multispectral radiometer mounted on an Autonomous Underwater Vehicle (AUV) operating under first year sea ice in Antarctica. AUVs provide a more stable platform for investigating spatial variability than Remotely Operated Vehicles (ROVs), which have been applied in recent years in similar conditions, as AUVs autonomously maintain standoff distance from the ice surface and the collected data is better able to be geo-located. Observed levels of patchiness of the ice algae assemblage varied between 21-66 m. This is well below the typical sampling regimes using sample coring techniques from the surface and is only possible using robotic platforms. While sea ice imposes logistical challenges on the use of such technologies, the development of sensor arrays and the platforms that carry them offer a way forward to better understand this most cryptic of sea ice algae communities. The development of such non-invasive techniques to measure environmental conditions under ice at scales commensurate with the inherent variability of these algal communities allows new understanding that would be otherwise impossible and represents a step change in our approach to studying these communities.

### Acknowledgements

Logistic support was provided by Antarctica New Zealand, and the research was made possible by primary funding from the New Zealand Antarctic Research Institute (NZARI-2014-3). In addition, the authors would like to thank the ongoing support of the Danish Council for Independent Research (Project DFF – 1323-00335, Sea ice ecosystems: Ecological effects of a thinning snow cover) and the Australian Research Council's Special Research Initiative for Antarctic Gateway Partnership (Project ID SR140300001). Griffiths University and the Defence Science and Technology Group (Australia) also contributed through equipment support. Final thanks goes to Rowan Frost (Australian Maritime College) and the K081 Team for all of their work to make this project a success.

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





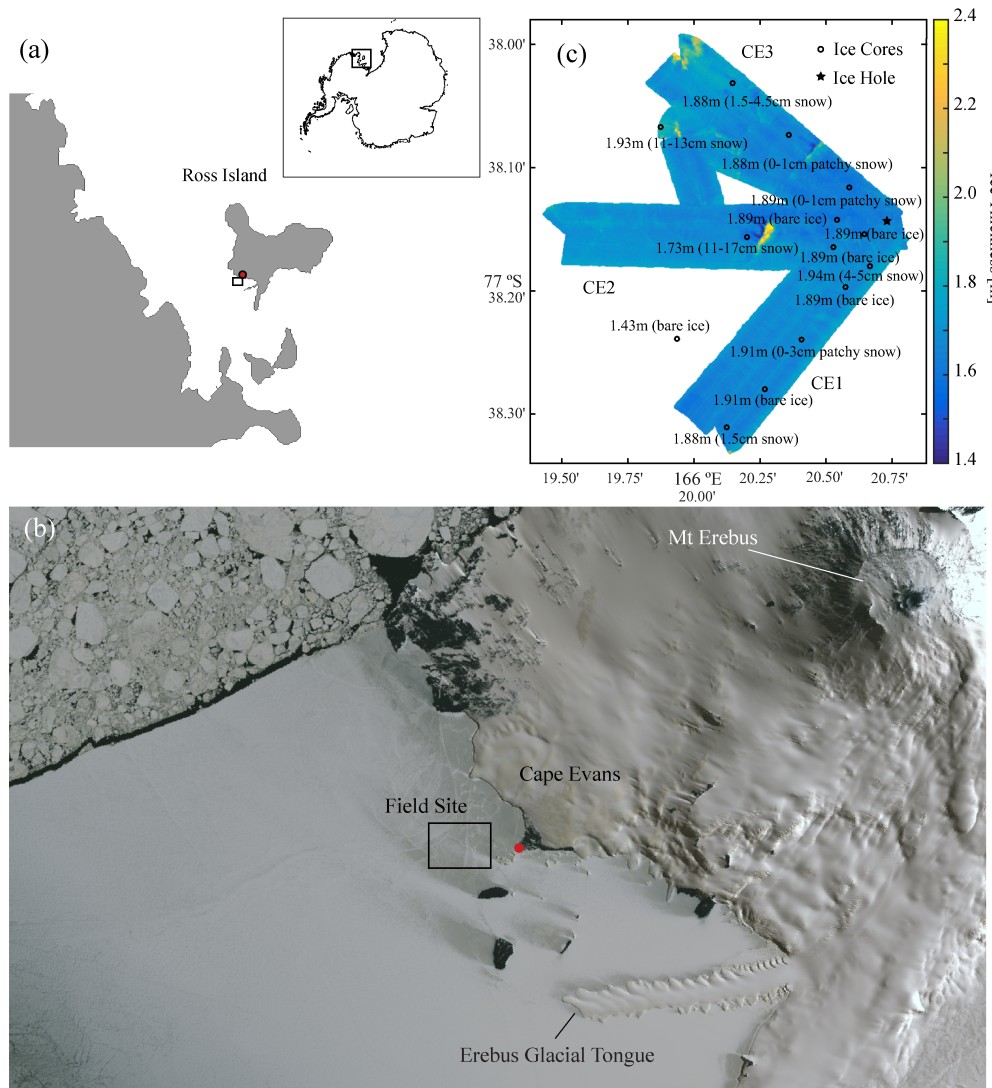

**Figure 1. (a) Location of Cape Evans, McMurdo Sound within the Antarctic continent (inset); (b) a WorldView-1 satellite image collected on 2 Nov 2014 showing the snow coverage in the region around Cape Evans; (c) Ice thickness as mapped using the interferometric Geoswath*plus* sonar onboard the AUV on the three transects where spectral intensity measurements were made (CE1, CE2 and CE3) across 60 m swaths with point measurements of ice thicknesses and snow condition measured at each of the coring stations (open circle) in addition to the ice hold (filled star) where the vehicle was deployed.**



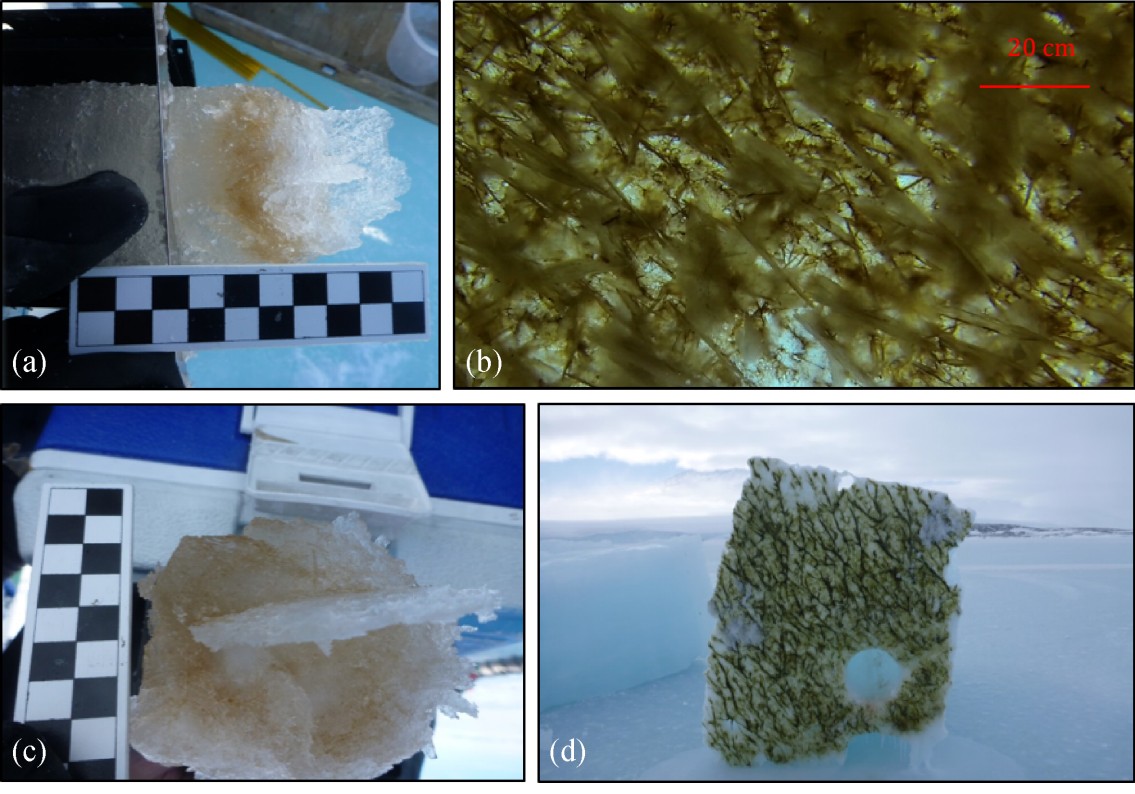

Figure 2. (a) The brown color towards the bottom of the sea ice core indicates ice algae (ice core widths 76.2 mm); (b) an underwater image of the ice-water boundary looking towards the bottom of the ice, demonstrating the ice regime and algal community complexity; (c) unconsolidated ice providing vertical surface area which ice algae can colonize; and, (d) a ~1x1 m, cross-sectional block of sea ice removed to allow access for the AUV.





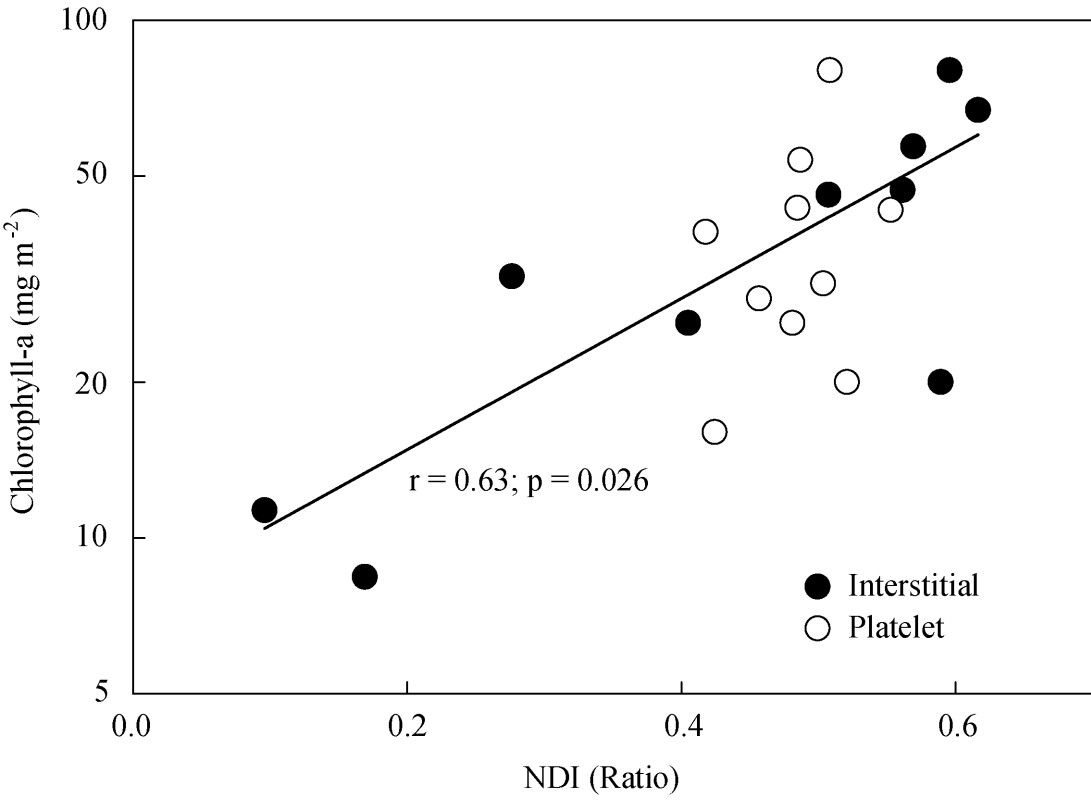

**Figure 3. Relationship between NDI (470:565) and chlorophyll concentration in a set of calibration samples (N = 19) from the same study area and time as the AUV transects. Each sample is either labeled as having algae present in the interstitial ice (filled circle) or in the platelet ice (clear circle). The line is fitted by least squares regression, and is of the form ln(chla) = 1.97-3.30 x NDI(471:565). The regression is significant at p = 0.026 and an adjusted $r^2$ of 0.403.**




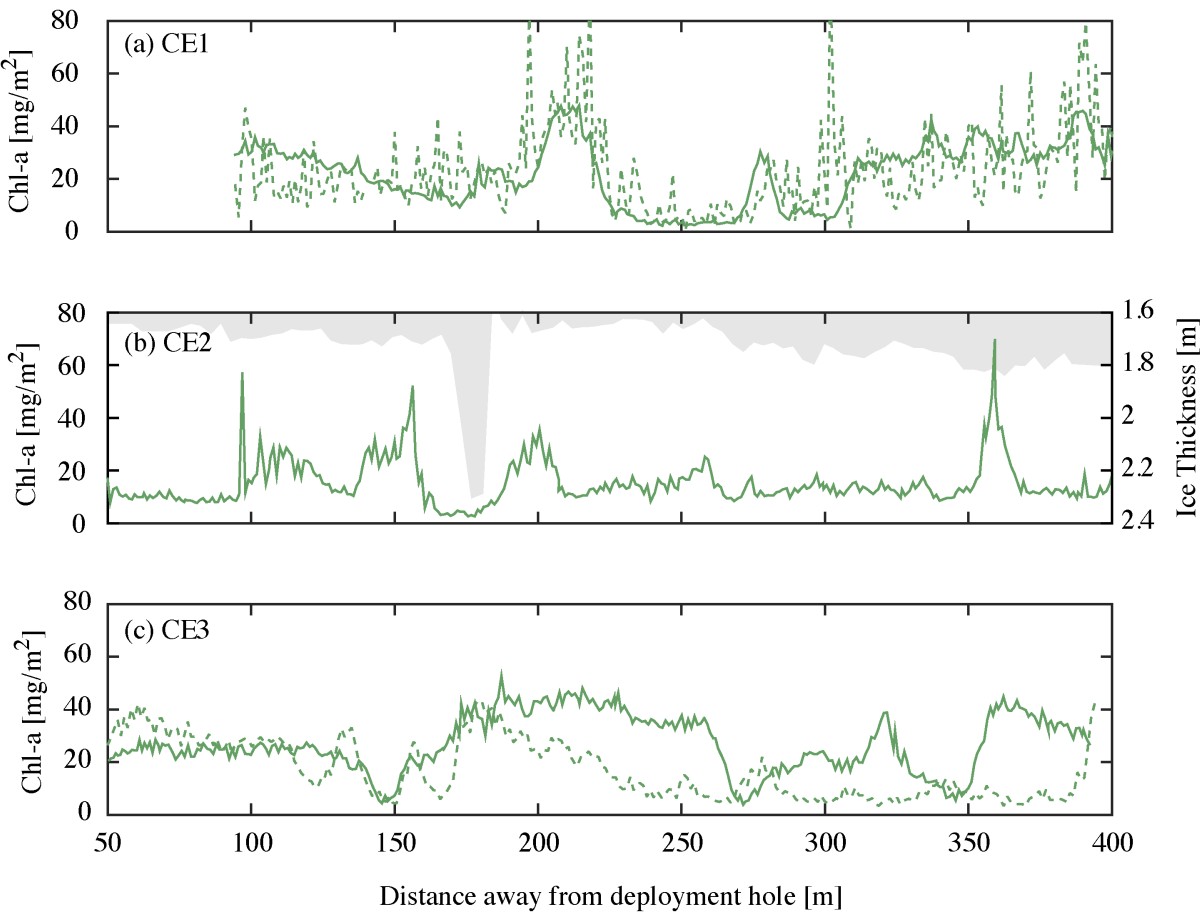

**Figure 4. (a) chla values for repeat runs of the outgoing transit leg of CE1 at local time 23:45 04 Nov 2014 (green dashed line) and 09:00 05 Nov 2014 (green solid line); (b) chla values for CE2 at local 10:45 05 Nov 2014 (green line) and overhead ice keel depth (bin averaged in 5 m intervals; greyed out) for the incoming transit leg; and (c) chla values for outgoing (green line) and incoming (green dashed line) transit legs (spaced at 20 m) for CE3 recorded at local time 09:45 05 Nov 2014.**