# Peer review of "Brief Communication: Capturing scales of spatial heterogeneity of Antarctic sea ice algae communities"

_The Cryosphere, 2016_

## Referee Comment (RC1) · Anonymous Referee #1 · 11 Oct 2016

This is an interesting study and the authors have collected a unique dataset using cutting edge methodology. The paper is generally well written and structured. However, in my opinion the paper has some shortcomings in regards to some data analyses and text, and I feel this unique dataset has not been utilized to its full extent. Below I have provided numerous remarks on the text as it is often vague and long-winded. In several instances I also suggested to cite more relevant and recent literature. Furthermore I made additional suggestions for more in-depth analyses of the data. Key critical points are a) the development of the NDI to chlorophyll a relationship (20 versus 60 data points, presentation of Figure 3 ), b) a lack of a discussion of effects of the distance (6m) of the AUV sensor to the subsurface of the ice (is there any information

on the chlorophyll concentration in the under-ice water, how did this distance affect signal/noise ratios and NDI-based chlorophyll a estimates) , c) lack of information on AUV location and potential navigation errors, and c) the calculation of patch-sizes based from data with a large foot-print (of the radiometer), e.g how were data-points for Fig.4 calculated. A revised manuscript might not fit the TC "Brief Communication" format anymore, however a detailed "Supplementary Information" section might be useful and could help to keep the manuscript in a short format. Given these shortcomings the manuscript requires major revisions.

Abstract: Please focus the abstract on your study and your results. In particular the last two sentence are vague. I would prefer to see some data on algal biomass, ice thickness and snow thickness from this study in the abstract, rather than a description of "where to go next". More generally, I suggest to focus the manuscript on the scientific results rather than on the innovation in engineering.

P1, L 12: "to quantifying ecosystem responses" (quite a long shot to go from simple observations to predictions...)

P1, L14:.. to describe the spatial heterogeneity of ice algal distribution.

P1, L15: rather define the ice as "land-fast sea ice" rather than first-year......

P1, L15-16: "These results....monitored" (Please clarify sentence)

P1, L 16 -17: This a very vague statement. The manuscript does not provide any information how ice algal communities could be categorized or how productivity could be measured. I suggest to delete and re-write abstract as per above comments.

Introduction: P1, L 23 -24: Please be more specific. Focus could be on the entire Southern Ocean (e.g. Saenz & Arrigo 2012, Arrigo 2014, Meiners et al. 2012) or Antarctic fast-ice ecosystems. Rysgaard is an older (and Arctic) reference. Maybe cite Mundy et al. 2007 and the recent papers of Campbell et al. 2014, 2015?

P1, 28 -29: This statement is true for Arctic sea ice and Antarctic land-fast sea ice, but

not necessarily true for "Antarctic sea ice" (see Horner et al. 1992, Arrigo 2014).

P1, 29-31: Please clarify sentence (Significance of what?)

P2, L1: Classic work on ice algal patchiness in Antarctic land-fast sea ice has been conducted by Swadling et al. (1997). This would be a good citation.

P2, L5: Maybe re-phrase? "Links between ice algal biomass and the under-ice light field are well established. . ..

P 2, L 9-10: As far as I understand NDI explain variability in the biomass (e.g. in a dataset used for calibration efforts), but a NDI does not explain spatial variability! Please correct this sentence.

I suggest major re-write of the introduction. It should provide an overview of a) the importance of Antarctic land-fast sea ice studies and b) recent advances in technologies to measure ice algal biomass non-invasively.

Methods: P2, L 13: This study did not measure transmittance! Transmittance is defined as the ratio between incoming solar radiation at the surface of a medium and the amount of radiation at the bottom of the medium. Rather than "spectral properties of light transmittance" the "multi-spectral under-ice irradiance" was measured. Please use proper terminology.

P2 L17-18: "patches a few meters across" and "very small (1s to 10s of meters" is duplicated information, I suggest to rewrite/combine these sentences

P2, L18: report snow thickness in "m" (SI unit) rather than "cm"

P2, Line 23: replace "Southern Ocean" with "land-fast sea ice in McMurdo Sound".

P 2, L25: Could you please provide a linear regression and an R2 for the ice thickness (measured from cores) versus ice thickness (measured from the sonar) relationship. This would greatly help to understand sonar-based ice thickness error.

P2, L30: "greater spatial precision", please provide information on the AUV positioning system and its accuracy, it might be useful to cite recent AUV ( Katlein et al. 2015, JGR) here

P3, L4-5: Given the speed of the vehicle and the downsampling to 1Hz there will still be some overlap of the area measured. Was this accounted for (e.g. through calculating a running mean?)

P3, L 10: Are there any measurements of the Chlorophyll a concentration in the under-ice water available? How does integrated Chla in in the 6m of water-column compare to the integrated Chla concentration in the overlying sea-ice? This would be very useful information to understand the signal:noise ratio in the measurements.

P3, L 28: "two volumes"? , please be more specific and maybe add a reference for this methodology

P4, L1: please provide reference for method of determination of chlorophyll a

P4, L13-15: Much more detail is required how the calibration of the AUV data was "tuned". It would be preferable to 1) show the results from all (60?) sites where ice cores and radiometer measurements were taken simultaneously and to discuss the derived best NDI (e.g. to compare with results from previous NDI studies) 2) show how these results were affected from discarding "selected" cores 3) show/explain the AUV calibration tuning method in detail and discuss potential error propagations through these 3 steps Also please explain NDI "(Ratio)" as x-axis descriptor in Figure 3. Is this indicating a further normalization of the originally determined NDI results, or should this read just "NDI" Figure caption indicates: N=19, but it appears that it is N=20.? Text indicates 14 sampling sites with a total of 60 "replicate" measurements Are these results from the 14 or 20 sites or is this a random subset of the 60 individual measurements. It appears that using the full 60 individual measurements would result in a stronger statistical relationship of NDI versus integrated algal biomass. "Damaged" core could be plotted in a specific colour. This would make are stronger argument for the used

NDI to chla relationship (which has a relatively poor fit).

Results: P4, L 21: do you mean "interstitial AND in the platelet ice", e.g. in figure 3 samples are distinguished between "interstitial" versus "platelets". Please clarify statement.

In addition algal communities in Figure 2b and 2d are looking different to those pictured in 2a and 2c. Algae in 2b and 2d closely resemble "stand" communities as commonly found in the Arctic. Were strands also evident in any of the core samples?

P4, Line 26: I understand that the NDI (470 nm : 565) was used for calibration of the multi-spectral AUV-based measurements but please also show the best NDI to Chla relationship that was derived from the 60 point measurements. This might help future studies in selecting the bands of AUV mounted radiometers.

P 4, L 26: Given that for NDI (470 nm : 565 nm) versus Chla the R2 was only 0.403, I doubt that this NDI explained the "highest proportion of the variability in algae biomass"

P5, L5: In this paragrapah I would just state that the poor predictive power of the algorithm was a result from the limited number wavelengths available from the AUV mounted radiometer.

Again it would be good to show the best NDI to Chla relationship as derived from the high-resolution/hyperspectral point measurements.

P 5, L 7: I suggest to delete reference to "multi-year sea ice was largely absent". Was there any multi-year sea ice in your sampling area of 500 x 500 ?

P4, Line 11: shorten "we consider it possible to assume" ?

Figure caption 1: ".….where spectral intensity measurements were made (CE1, CE2 and CE3) across 60 m swaths", please specify if this statement is correct. In the Methods sections it is explained that that the footprint of the radiometric measurements was about 4m in diameter. This is contradicting 60 m wide swaths.

Figure 3: I suggest to show data with a log scale y-axis starting at 1 and ending at 100 mg Chla m-2. The current y-axis starting at 5 and with a maximum of 100, and at the same time showing "linear" axis-descriptors is confusing. Why are only 20 data points shown, when you have measured 60? It would be preferable to see all 60 paired measurements of Chla versus NDI.

P5, L 17-18: Clarify sentence there seems to be an "and" missing.

Discussion: P 5, L 23-24: No data on the varying amplitude and length scales of patchiness for spectral irradiance are shown. Please be more specific in the terminology used to describe your data.

P5, L 23-24: "For each transect, chla estimates were interpolated to constant spacing between samples to allow a spectral analysis of the data using a standard Fast Fourier Transform (FFT) similar to time series analysis". This information needs to go in the Methods part of the manuscript. Define "constant spacing". Is this the length scale of the radiometric foot-print of approx. 4 m in diameter? Please detail you produced the data presented in Figure 4. They seems to have a higher resolution than the maximum resolution that can be achieved by the measurements, e.g. the 4 m sensor footprint.

P5, L 26-27: Please show these correlograms (maybe in a Supplementary Information section?) .Variograms have been calculated for light transmission in Arctic sea ice and this would allow for interesting comparisons (Katlein et al. 2015).

P6, Line 3: There are more relevant references available for the spatial variability of ice algal biomass, e.g. Rysgaard et al. 2001, Steffens et al. 2006: Søgaard et al. 2013). I suggest to cite these and discuss new data (this study) in relation to these previous studies.

P 5, Line 10: Here it would be good to cite and relate to some other studies that investigated snow – ice algal biomass relationships. One additional sentence about potential thresholds in the snow ice algal relationship and the seasonally changing

influence of snow on ice algae would be useful.

P5, Line 15: One would assume that most possibly platelet ice – (partly) consolidated into the ice sheet was driving the biomass distribution. From Figure 3 it appears that platelet ice was associated with higher algal biomass. Could the sonar data shed some light into this? E.g. could the sonar data be used to detect platelet ice patches – presumably associated with a higher surface roughness? This study provides a unique dataset for testing this. Rather than mentioning this as a hypothesis, I suggest to use the data to test this relationship.

P6, L 7: Ate there any ice draft data available from the other two transects? Why were analyses of the correlation between ice draft and NDI-derived algal distribution restricted to this single transect?

P 6, L 17-22: Please clarify both these sentences. Do you want to say: "The poor NDI to Chla relationship does not allow for more complex analyses of the relationships between physical sea ice properties and ice algal biomass. Nevertheless, our data provide proof-of-concept to use AUV technology to measure ice algal spatial variability"?

P6, Line 24: please define "infiltration ice" ?

P7, L3: "fast ice" rather than "first-year sea ice"

P7, L6: "well above" rather than "well below"? One could argue that most ice core studies work on areas of 10m by 10m .

P7, L 7: "ice coring" rather than "sample coring"

P7, 9: "this most cryptic of sea ice algae communities"? most cryptic compared to what other communities?

Figure 4: a) Why is sea ice draft only shown for one transect, when data for all 3 transects are available ( see Fig.1) b) What is the bin-size for the chlorophyll a lines. It appears that they are shown at a higher resolution than 4 m, but in the text 4 m (radius)
is given as the radiometric foot-print of the sensor. Please explain and adjust (bin) chla values to 4 m length scale - if necessary.

---

## Referee Comment (RC2) · Anonymous Referee #2 · 28 Oct 2016

Overall the information presented represents valuable information regarding the feasibility of using optics to infer algal biomass in sea ice and the horizontal variability of such in land fast ice of Antarctica. That said, there are several areas in the ms that deserve improvements. Among the areas that need improving is work to provide better citations that will help place the current work in better context of the scientific progress over the past 20+ years. Additionally better citations will help the authors to provide better and more accurate information regarding sea ice and sea ice biota. Specific examples are given below.

Page 1: First sentence of abstract (line 12) is hyperbole and not neededline 23- There are much better references than just one reference from McMinn et al

1999 for the assertions- including review papers by working groups.

line 28- is an assertion that is simply not true- the idea that ice algae predominantly grow at the bottom of the sea ice is a fallacy propagated by thinking that McMurdo sound sea ice is representative of 20M sq kilometers of sea ice- when in fact ice algae grow throughout all sorts of areas of the sea ice in the pack ice regimes. Moreover, fall blooms are also common- refer to the work by Garrison and others as well the Japanese authors on the topic.

Page 2: line 5- again only one reference is not sufficient - e.g. citing Palmisano's work would be appropriate

Page 3- line 24- Sullivan et al's prior work in the 80's also quantified the percentage of the biomass in different layers of the bottom ice.

Page 4 line 13- 14- the sentence is very hard to translate. How did they scale the estimates? what does this mean? was there a correction factor applied and if so, how?

Page 5- line 6- referencing something that may or may not be published somewhere else does not seem like it should be allowed.

line 10 - same issue- it seems relevant to present the data herin.

Page 6 line 7 - the sentence is very awkward- and meaning is obscured. suggest re-writing to clarify.

Page 7- line 2- the term pioneering seems a bit much as this does not seem like pioneering work. There has been much work on this topic and approach already. More appropriately the incremental work demonstrates ability to use place optical instrumentation on underwater vehicles to try to estimate biomass over larger spatial scales.

line 10- assertion that new understanding is possible- would be more convincing if the authors presented on this manuscript what new understanding they have provided by doing the exercise. The ms does not convince me that they have contributed more

understanding yet. perhaps if they did a rigorous spatial analysis of their transects to inform us of the spatial scales of patchiness or autocorrelation THEN i could see that they might be providing better and new information for better understanding sea ice and sea ice algae.

Figure 1- the orientation of Antarctica in the inset looks transposed. The text on the transect figure inset make this figure problematic- it is not readable

Figure 4- it would be nice to add a panel that shows the depth of the radiometers on the vehicle as it made the transect.

Fig 4c- does the repeat of the lines imply there is a lack of repeatability ? is there an issue being hid hear concerning the stability of the incoming surface radiation during the measurements?

again- this seems like a nice data set to work with- but there are issues to be addressed in the analyses and the presentation that would make the publication much better.

---

## Editor Comment (EC1) · J.-L. Tison (Editor) · 30 Jan 2017

Dear Authors,

Thanks for posting your replies to reviewers comments. I plan to read these next Wednesday (08 Feb) and return to you on that day (hopefully) with advise concerning the potential production of a final manuscript.

Best regards,

Jean-Louis Tison

---

## Author Comment (AC1) · 30 Jan 2017

General Comments from the Reviewers

General Comments from Reviewer 1

**Comment:** *This is an interesting study and the authors have collected a unique dataset using cutting edge methodology. The paper is generally well written and structured. However, in my opinion the paper has some shortcomings in regards to some data analyses and text, and I feel this unique dataset has not been utilized to its full extent. Below I have provided numerous remarks on the text as it is often vague and long-winded. In several instances I also suggested to cite more relevant and recent literature. Furthermore I made additional suggestions for more in-depth analyses of the data. Key critical points are a) the development of the NDI to chlorophyll a relationship (20 versus 60 data points, presentation of Figure 3), b) a lack of a discussion of effects of the distance (6m) of the AUV sensor to the subsurface of the ice (is there any information on the chlorophyll concentration in the under-ice water, how did this distance affect signal/ noise ratios and NDI-based chlorophyll a estimates), c) lack of information on AUV location and potential navigation errors, and c) the calculation of patch-sizes based from data with a large foot-print (of the radiometer), e.g how were data-points for Fig.4 calculated. A revised manuscript might not fit the TC "Brief Communication" format anymore, however a detailed "Supplementary Information" section might be useful and could help to keep the manuscript in a short format. Given these shortcomings the manuscript requires major revisions.*

**Response:** The authors would like to thank the Reviewer for their comments. Care has been taken to improve the work and address their concerns as per the specific comments below. As noted by the Reviewer, the authors feel that the length restraint of a Brief Communication prohibits the inclusion of a Supplementary Information section.

Specific Comments from Reviewer 1

| Reviewer Comments | Reply |
|---|---|
| Abstract: Please focus the abstract on your study and your results. In particular the last two sentence are vague. I would prefer to see some data on algal biomass, ice thickness and snow thickness from this study in the abstract, rather than a description of "where to go next". More generally, I suggest to focus the manuscript on the scientific results rather than on the innovation in engineering. | Efforts have now been made to address this concern while still remaining within the word limit. The balance that the authors were aiming for is to highlight the accomplishments in both the science and the engineering. |
| P1, L 12: "to quantifying ecosystem responses" (quite a long shot to go from simple observations to predictions: : :) | The aim of this opening sentence is to highlight the importance of understanding spatial variability of algal communities. Although the authors aren't suggesting that predictions be made here, it is maintained that there is an importance in understanding this variability if attempts to model the community are to be made. |
| P1, L14:.. to describe the spatial heterogeneity of ice algal distribution. | This has now been amended in the revised abstract. |
| P1, L15: rather define the ice as "land-fast sea ice" rather than first-year, | Abstract has now been amended. |
| P1, L15-16: "These results...monitored" (Please clarify sentence) | Abstract has now been amended. |
| P1, L 16 -17: This a very vague statement. The manuscript does not provide any information how ice algal communities could be categorized or | Abstract has now been amended. |

| | |
|---|---|
| how productivity could be measured. I suggest to delete and re-write abstract as per above comments. | |
| Introduction: P1, L 23 -24: Please be more specific. Focus could be on the entire Southern Ocean (e.g. Saenz & Arrigo 2012, Arrigo 2014, Meiners et al. 2012) or Antarctic fast-ice ecosystems. Rysgaard is an older (and Arctic) reference. Maybe cite Mundy et al. 2007 and the recent papers of Campbell et al. 2014, 2015? | Citations have now been updated to better reflect the material. |
| P1, 28 -29: This statement is true for Arctic sea ice and Antarctic land-fast sea ice, but not necessarily true for "Antarctic sea ice" (see Horner et al. 1992, Arrigo 2014). | Amended to now read: *'Sea ice algae in McMurdo Sound, Antarctica comprise a shade-adapted autotrophic assemblage…'* |
| P1, 29-31: Please clarify sentence (Significance of what?) | Amended to now read: *'The significance of this algal community to marine ecosystems as a concentrated food resource prior to ice break-up, and its potential role in seeding pelagic production subsequent to ice break-up, are widely recognized (Arrigo & Thomas, 2004).'* |
| P2, L1: Classic work on ice algal patchiness in Antarctic land-fast sea ice has been conducted by Swadling et al. (1997). This would be a good citation. | Citation has now been added and the authors would like to thank the reviewer for this suggestion. |
| P2, L5: Maybe re-phrase? "Links between ice algal biomass and the under-ice light field are well established… | The phrasing here has now been kept the same but the citations have been updated to reflect more current work and the expansion into remote sensing techniques. |
| P 2, L 9-10: As far as I understand NDI explain variability in the biomass (e.g. in a dataset used for calibration efforts), but a NDI does not explain spatial variability! Please correct this sentence. | This sentence has now been slightly modified by removing the word 'spatial'. |
| I suggest major re-write of the introduction. It should provide an overview of a) the importance of Antarctic land-fast sea ice studies and b) recent advances in technologies to measure ice algal biomass non-invasively. | Changes have been made to the Introduction to improve the flow but structural the authors feel that it was important to maintain the key ideas of ice algae / patchiness / NDI as this links to the rest of the paper. As for new, emerging technologies for non-invasive sampling, this has been left for the Discussion as there have been so few studies to date. |
| Methods: P2, L 13: This study did not measure transmittance! Transmittance is defined as the ratio between incoming solar radiation at the surface of a medium and the amount of radiation at the bottom of the medium. Rather than "spectral properties of light transmittance" the "multi-spectral under-ice irradiance" was measured. Please use proper terminology. | This has now been corrected as suggested. |
| P2 L17-18: "patches a few meters across" and "very small (1s to 10s of meters" is duplicated information, I suggest to rewrite/combine these sentences | This has now been corrected as suggested. |
| P2, L18: report snow thickness in "m" | This has now been corrected as suggested. |

| | |
|---|---|
| (SI unit) rather than "cm" | |
| P2, Line 23: replace "Southern Ocean" with "land-fast sea ice in McMurdo Sound". | This has now been corrected as suggested. |
| P 2, L25: Could you please provide a linear regression and an R2 for the ice thickness (measured from cores) versus ice thickness (measured from the sonar) relationship. This would greatly help to understand sonar-based ice thickness error. | A more explicit validation of this technique using these datasets has just been published in Lucieer et al. (2016).

*Lucieer, V., Nau, A. W., Forrest, A. L., & Hawes, I.: Fine-Scale Sea Ice Structure Characterized Using Underwater Acoustic Methods. Remote Sensing, 8(10), 821, doi:10.3390/rs8100821, 2016.*

This reference and the mean difference of 0.11 m between measurement techniques has now been included. |
| P2, L30: "greater spatial precision", please provide information on the AUV positioning system and its accuracy, it might be useful to cite recent AUV (Katlein et al. 2015, JGR) here | This has now been corrected as suggested. |
| P3, L4-5: Given the speed of the vehicle and the downsampling to 1Hz there will still be some overlap of the area measured. Was this accounted for (e.g. through calculating a running mean?) | While there will be some overlap at 1Hz, the data are presented as point values rather than estimating a running mean. All this would do would be to further smooth the signal and so wasn't presented here. |
| P3, L 10: Are there any measurements of the Chlorophyll a concentration in the underice water available? How does integrated Chla in in the 6m of water-column compare to the integrated Chla concentration in the overlying sea-ice? This would be very useful information to understand the signal:noise ratio in the measurements. | No direct measurements of Chla concentrations were made in the water; however, in addition to the Satlantic OCR507 multispectral radiometer, the AUV was also making estimates of Chla using a Wetlabs Ecopuck sampling at 4Hz. The values recorded were at the minimum detection limit of the instrument and so it was assumed that the concentrations in the water column were minimal. However, in order to address the problem of attenuation through the water column, a TriOS Ramses ACC VIS cosine-corrected hyperspectral radiometer was used to derive attenuation coefficients for the different wavelengths. These profiles were then used to correct for water column attenuation as detailed in text to address the question of the influence of the water column on the observed results:

*'To account for variable AUV depth and ice cover thickness, the spectral irradiance at each wavelength was calculated for the ice-water interface using a vertical exponential attenuation model. This combined wavelength-specific attenuation coefficients for transmitted irradiance (Kirk, 2011), measured on site, with AUV depth below the underside of the ice estimated from sonar measurements of overhead ice draft.'* |
| P3, L 28: "two volumes"? , please be more specific and maybe add a reference for this methodology | This has now been amended to read *'adding two times the volume of melted seawater'* as per the protocol established by Rintala et al. (2014) [reference added]. |
| P4, L1: please provide reference for method of determination of chlorophyll a | Reference has now been added. |
| P4, L13-15: Much more detail is required how the calibration of the AUV data was "tuned". It would be preferable to 1) show the results from all (60?) sites where ice cores and radiometer measurements were taken simultaneously and to discuss the derived best NDI (e.g. to compare with | For clarification, there were 20 stations where ice cores were collected in triplicate (i.e., three samples with a 1.2 m diameter). As these samples were so close together, it wasn't possible to use individual measurements for the NDI calculation using the Satlantic signal as the footprint of the Satlantic was 4 m in diameter, and there was often some considerable variability in chla content of "replicate" cores at each station. For these reasons, the TriOS data were used to develop the NDI, and we used the average results of the triplicate sites (note that N=19 in the caption for |

| | |
|---|---|
| results from previous NDI studies) 2) show how these results were affected from discarding "selected" cores 3) show/explain the AUV calibration tuning method in detail and discuss potential error propagations through these 3 steps Also please explain NDI "(Ratio)" as x-axis descriptor in Figure 3. Is this indicating a further normalization of the originally determined NDI results, or should this read just "NDI" Figure caption indicates: N=19, but it appears that it is N=20.? Text indicates 14 sampling sites with a total of 60 "replicate" measurements Are these results from the 14 or 20 sites or is this a random subset of the 60 individual measurements. It appears that using the full 60 individual measurements would result in a stronger statistical relationship of NDI versus integrated algal biomass. "Damaged" core could be plotted in a specific colour. This would make are stronger argument for the used NDI to chla relationship (which has a relatively poor fit). | Figure 3 has now been amended to N=20 as detailed in the text) to integrate over a larger footprint area given the small-scale variability that was apparent in the core data. The "tuning" of the NDI that the reviewer refers to was undertaken as the geometric mean of chl-a concentrations derived from applying the TriOS-derived NDI to the Satlantic data was 18.7 compared to 26.7 from the measured ice cores. The reason for this is not clear, but we considered likely to result from optical measurement differences in the band-pass instrument (Satlantic) and the higher resolution hyperspectral (TriOS) instrument.

The references to discarding cores refers to one of the three replicates at two different stations. Efforts were made in the field to capture the bottom fragments by including pieces from within the hole. However, chla values were substantially lower than in some replicates and so were thus discarded from the analysis. The following line has now been included to provide clarity: *'Of the 60 cores total that were collected, only one replicate from 2 different stations were so damaged.'*

Finally, both the caption and the axes label of Figure 3 has been amended to only read NDI to avoid any confusion as there was no further normalization. This was originally included to indicate a variable without units. |
| Results: P4, L 21: do you mean "interstitial AND in the platelet ice", e.g. in figure 3 samples are distinguished between "interstitial" versus "platelets". Please clarify statement. | This has now been corrected as suggested. |
| In addition, algal communities in Figure 2b and 2d are looking different to those pictured in 2a and 2c. Algae in 2b and 2d closely resemble "stand" communities as commonly found in the Arctic. Were strands also evident in any of the core samples? | While the authors agree with the reviewer that there are similarities with strand communities, and there was attachment of algae on platelet ice (e.g. Figure 2b), strands were not observed in any of the cores. |
| P4, Line 26: I understand that the NDI (470 nm : 565) was used for calibration of the multi-spectral AUV-based measurements but please also show the best NDI to Chla relationship that was derived from the 60 point measurements. This might help future studies in selecting the bands of AUV mounted radiometers. | This point has been discussed in reply to the previous comment by the reviewer. |
| P 4, L 26: Given that for NDI (470 nm : 565 nm) versus Chla the R2 was only 0.403, I doubt that this NDI explained the "highest proportion of the variability in algae biomass" | While the authors agree that the R2 is low, the intent of this sentence was to say that this NDI explained the highest proportion of the variability in algae biomass *results*, not in a global context. To provide clarification, 'results' has been added to the sentence. |
| P5, L5: In this paragraph I would just state that the poor predictive power of the algorithm was a result from the limited number wavelengths available from the AUV mounted radiometer. | This paragraph has now been amended to reflect both possible reasons why this might be the case. |
| Again it would be good to show the | The authors are unsure what to address here as the best results have been |

| | |
|---|---|
| best NDI to Chla relationship as derived from the high-resolution/hyperspectral point measurements. | presented in this work already. |
| P 5, L 7: I suggest to delete reference to "multi-year sea ice was largely absent". Was there any multi-year sea ice in your sampling area of 500 x 500 ? | The reference to multi-year sea ice has been removed although the sentence was left in there to describe the snow conditions. |
| P4, Line 11: shorten "we consider it possible to assume" ? | This has now been amended to 'we assume'. |
| Figure caption 1: "...where spectral intensity measurements were made (CE1, CE2 and CE3) across 60 m swaths", please specify if this statement is correct. In the Methods sections it is explained that that the footprint of the radiometric measurements was about 4m in diameter. This is contradicting 60 m wide swaths. | In rereading this caption, it is understood how the description of the sonar swath (60 m) could be misleading to the reader. The caption has now been amended. |
| Figure 3: I suggest to show data with a log scale y-axis starting at 1 and ending at 100 mg Chla m-2. The current y-axis starting at 5 and with a maximum of 100, and at the same time showing "linear" axis-descriptors is confusing. Why are only 20 data points shown, when you have measured 60? It would be preferable to see all 60 paired measurements of Chla versus NDI. | The authors feel that 60 cores collected at the 20 stations has been explained previously and has also been clarified in the text.

The limits of the y-axis of Figure 3 has now been amended to go from 1 to 100 for better clarity. |
| P5, L 17-18: Clarify sentence there seems to be an "and" missing. | This has now been corrected as suggested. |
| Discussion: P 5, L 23-24: No data on the varying amplitude and length scales of patchiness for spectral irradiance are shown. Please be more specific in the terminology used to describe your data. | This sentence has now been amended to be more exact. |
| P5, L 23-24: "For each transect, chla estimates were interpolated to constant spacing between samples to allow a spectral analysis of the data using a standard Fast Fourier Transform (FFT) similar to time series analysis". This information needs to go in the Methods part of the manuscript. Define "constant spacing". Is this the length scale of the radiometric foot-print of approx. 4 m in diameter? Please detail you produced the data presented in Figure 4. They seems to have a higher resolution than the maximum resolution that can be achieved by the measurements, e.g. the 4 m sensor footprint. | This has now been amended. The data reported in Figure 4 is at a sampling frequency of 1Hz (i.e. the raw data) and so is not the data that is then smoothed at a 1 m constant interval setting to conduct the Fast Fourier Transform. This is because FFTs require a constant spacing that isn't possible when the AUV has irregular surge. |
| P5, L 26-27: Please show these correlograms (maybe in a | The authors have chosen not to include correlograms in this very short communication. |

| | |
|---|---|
| Supplementary Information section?). Variograms have been calculated for light transmission in Arctic sea ice and this would allow for interesting comparisons (Katlein et al. 2015). | |
| P6, Line 3: There are more relevant references available for the spatial variability of ice algal biomass, e.g. Rysgaard et al. 2001, Steffens et al. 2006: Søgaard et al. 2013). I suggest to cite these and discuss new data (this study) in relation to these previous studies. | References to earlier work by Rysgaard et al. (2001), Gosselin et al. (1986) and Swadling et al. (1997) have now all been included. While these other authors have identified a wide variation in scales, the advantage of our current work is that it is unique in that the transect lines provide continuous lines of coverage from which the length scales of variance can be estimated. The text has been amended slightly to provide greater clarity. |
| P 6, Line 10: Here it would be good to cite and relate to some other studies that investigated snow – ice algal biomass relationships. One additional sentence about potential thresholds in the snow ice algal relationship and the seasonally changing influence of snow on ice algae would be useful. | A citation for this has now been added and the potential thresholds in the snow ice algal relationship has been left for other discussions in the interest of space in a Short Communication as well as the fact that very minimal snow existed in the region. |
| P6, Line 15: One would assume that most possibly platelet ice – (partly) consolidated into the ice sheet was driving the biomass distribution. From Figure 3 it appears that platelet ice was associated with higher algal biomass. Could the sonar data shed some light into this? E.g. could the sonar data be used to detect platelet ice patches – presumably associated with a higher surface roughness? This study provides a unique dataset for testing this. Rather than mentioning this as a hypothesis, I suggest to use the data to test this relationship. | The reviewer makes a good point here that has been explored in some detail by Lucieer et al. (2016). However, determining platelet structure from the sonar measurements hasn't been developed enough as a tool to develop correlation length scales. This is the focus of future work that is already currently being explored. |
| P6, L 7: Are there any ice draft data available from the other two transects? Why were analyses of the correlation between ice draft and NDI-derived algal distribution restricted to this single transect? | The reason for this transect being selected is that it had the greatest degree of variability in the ice structure and was meant to be the greatest example of this. The other two transects were essentially level ice and so weren't provided. |
| P 6, L 17-22: Please clarify both these sentences. Do you want to say: "The poor NDI to Chla relationship does not allow for more complex analyses of the relationships between physical sea ice properties and ice algal biomass. Nevertheless, our data provide proof-of-concept to use AUV technology to measure ice algal spatial variability"? | The text has now been amended to provide better clarification. |
| P6, Line 24: please define "infiltration ice" ? | This should have read 'interstitial ice' and has now been amended. |
| P7, L3: "fast ice" rather than "first-year sea ice" | This has now been corrected as suggested. |
| P7, L6: "well above" rather than "well below"? One could argue that most ice core studies work on areas of 10m by | This has now been corrected as suggested. |

| | |
|---|---|
| 10m. | |
| P7, L 7: "ice coring" rather than "sample coring" | This has now been corrected as suggested. |
| P7, 9: "this most cryptic of sea ice algae communities"? most cryptic compared to what other communities? | This has now been amended to avoid ambiguity. |
| Figure 4: a) Why is sea ice draft only shown for one transect, when data for all 3 transects are available ( see Fig.1) b) What is the bin-size for the chlorophyll a lines. It appears that they are shown at a higher resolution than 4 m, but in the text 4 m (radius) is given as the radiometric foot-print of the sensor. Please explain and adjust (bin) chla values to 4 m length scale - if necessary. | This comment has been previously addressed. |

**General Comments from Reviewer 2**

**Comment:** *Overall the information presented represents valuable information regarding the feasibility of using optics to infer algal biomass in sea ice and the horizontal variability of such in land fast ice of Antarctica. That said, there are several areas in the ms that deserve improvements. Among the areas that need improving is work to provide better citations that will help place the current work in better context of the scientific progress over the past 20+ years. Additionally, better citations will help the authors to provide better and more accurate information regarding sea ice and sea ice biota. Specific examples are given below.*

**Author Response:** The authors would like to thank the Reviewer for their comments. Care has been taken to improve the work and address their concerns as per the specific comments below.

**Specific Comments from Reviewer 2**

| Reviewer Comments | Reply |
|---|---|
| Page 1: First sentence of abstract (line 12) is hyperbole and not needed. | The abstract has been amended significantly in reflection from the first reviewer; however, the first sentence remains relatively unchanged. The authors feel that this is important to place the study in context. |
| Line 23- There are much better references than just one reference from McMinn et al 1999 for the assertions- including review papers by working groups. | This has now been corrected as suggested. |
| Line 28 - is an assertion that is simply not true- the idea that ice algae predominantly grow at the bottom of the sea ice is a fallacy propagated by thinking that McMurdo sound sea ice is representative of 20M sq kilometers of sea ice- when in fact ice algae grow throughout all sorts of areas of the sea ice in the pack ice regimes. Moreover, fall blooms are also common- refer to the work by Garrison and others as well the Japanese authors on the topic. | This sentence has now been amended to be specific about McMurdo Sound rather than being overly general. |
| Page 2: line 5- again only one reference is not sufficient - e.g. citing Palmisano's work would be appropriate. | This has now been corrected as suggested. |

| | |
|---|---|
| Page 3- line 24- Sullivan et al's prior work in the 80's also quantified the percentage of the biomass in different layers of the bottom ice. | While it is true that the work of Sullivan is important, the authors feel that this is addressed by referencing the work of Ryan et al. (2006) and Carnet et al. (2014). |
| Page 4 line 13- 14- the sentence is very hard to translate. How did they scale the estimates? what does this mean? was there a correction factor applied and if so, how? | This question was also addressed by the other reviewer and, similar to the previous response, is addressed in the text:

'To account for variable AUV depth and ice cover thickness, the spectral irradiance at each wavelength was calculated for the ice-water interface using a vertical exponential attenuation model. This model combined wavelength-specific attenuation coefficients for transmitted irradiance (Kirk, 2011), measured on site, with AUV depth below the underside of the ice estimated from sonar measurements of overhead ice draft.' |
| Page 5- line 6- referencing something that may or may not be published somewhere else does not seem like it should be allowed. | The text has now been modified to remove any ambiguity with other manuscripts being developed. |
| line 10 - same issue- it seems relevant to present the data herein. | The text has now been modified to remove any ambiguity with other manuscripts being developed. |
| Page 6 line 7 - the sentence is very awkward- and meaning is obscured. Suggest re-writing to clarify. | This sentence has now been amended for better clarity. |
| Page 7- line 2- the term pioneering seems a bit much as this does not seem like pioneering work. There has been much work on this topic and approach already. More appropriately the incremental work demonstrates ability to use place optical instrumentation on underwater vehicles to try to estimate biomass over larger spatial scales. | While the authors still feel that the use of AUVs for this application is pioneering, the text has been amended. |
| line 10- assertion that new understanding is possible- would be more convincing if the authors presented on this manuscript what new understanding they have provided by doing the exercise. The ms does not convince me that they have contributed more understanding yet. Perhaps if they did a rigorous spatial analysis of their transects to inform us of the spatial scales of patchiness or autocorrelation THEN I could see that they might be providing better and new information for better understanding sea ice and sea ice algae. | The authors feel that this was the aim in presenting both the correlograms as well as the FFT analysis of the spatial data (i.e. first paragraph of the discussion). |
| Figure 1- the orientation of Antarctica in the inset looks transposed. The text on the transect figure inset make this figure problematic- it is not readable | The orientation of this has now been fixed and the depth of snow coverage has been deleted as it is minimal. This is also discussed in detail in Lucieer et al. (2016). |
| Figure 4- it would be nice to add a panel that shows the depth of the radiometers on the vehicle as it made the transect. | As the vehicle operates at a fixed hydrostatic depth (6 m; first paragraph of the sampling design) it was decided not to add this panel in as it was felt that it didn't add much to the discussion. |
| Fig 4c- does the repeat of the lines imply there is a lack of repeatability ? is there an issue being hid hear concerning the stability of the | The purpose of the three subplots was to show a different hypothesis in each: 1) that the results were reproducible at different times of the day over the same transect; 2) that the observations made weren't related to the overhead ice draft; and, 3) that the results between transects were |

| incoming surface radiation during the measurements? | similar but not exactly the same. The aim was to examine this data in each possible testable hypothesis. |
|---|---|

---

## Editor Comment (EC2) · J.-L. Tison (Editor) · 16 Feb 2017

Dear Authors,

I have now received the appropriate versions of your "Authors responses" to the reviewers and new annoted manuscript, Thank you.

As mentioned earlier on, I will follow the request from one of the reviewers to have a look at this new version of the manuscript (and your authors comments).

I will keep you posted of progress,

Best regards,

[Figure]

Jean-Louis Tison

---

## Editor Comment (EC3) · J.-L. Tison (Editor) · 29 Mar 2017

Dear Alexander and co-authors,

As I mentioned earlier on, I had to ask the opinion of a "fresher" new referee on the new version of the paper, given the second feedback of one of the former reviewers. This new referee asked me to post his review to you, and here it is, attached to this comment. I will state my final decision now, in a separate post.

Cheers,

Jean-Louis

[Figure]

Please also note the supplement to this comment:
http://www.the-cryosphere-discuss.net/tc-2016-186/tc-2016-186-EC3-supplement.pdf

[Figure]

**Supplement:**

Overall comments:

The manuscript by Forrest et al. applies AUV-based measurements of transmitted multispectral irradiance to estimate spatial variability in bottom ice algal pigment biomass using a normalized difference index of 470 and 565 nm. The manuscript presents a very neat dataset of NDI-derived chlorophyll concentrations over three 500-m transects. However, there is a lot more that could have been accomplished and thus this submission seems to be a presentation of a partial dataset and analysis, and does not really have the impact to warrant a short note style publication. There could have been a lot more in-depth analysis of the spectral response of transmitted irradiance to the different cover types (e.g., snow, ice, chlorophyll, water, etc.). For example, analysis of the TriOS dataset could have been used to examine the hyperspectral response, which would provide arguments to support/refute limitations/benefits of the AUV-based multispectral sensor. This suggestion leads to an aspect of the manuscript that was troubling, i.e., the choice of multispectral wavelengths for the Satlantic sensor used on the AUV. Knowing that previous research had shown NDIs using wavelengths closer together (on the order of 10-20 nm apart) better explain ice algal pigment biomass variability, why were wavelengths chosen to be spaced far apart? That is, was there no choice on the wavelengths used in the sensor when originally purchased?

Another aspect of the manuscript that is perhaps the most troubling was the fact that a calibration of the AUV-NDI method is attempted, but in the end not used. Instead a range of core-based measurements is used to confine the NDI-values. Such a standardization is not really appropriate as it suggests NDI-estimates of chlorophyll concentration were absolute. A more simple approach could be to present un-calibrated NDI values as really, only spatial variance is analyzed in the AUV dataset. To support such a data presentation and analysis, an additional examination on the influence of snow depth, ice thickness and chl concentration on NDI values would have been useful to establish the dominant role of chl on NDI variability. Such an analysis would place greater confidence on similar statements made in the manuscript.

To conclude, I believe the dataset warrants publications and the pros/cons of the application of AUVs to estimate large spatial scale estimates of ice algal biomass be discussed in the general scientific audience. However, the presented analysis is largely incomplete and requires a more in-depth analysis and data presentation. Therefore, I suggest the manuscript is rejected, but potentially invited back for a full-length paper after a more complete analysis and interpretation is attained.

Line-by-line comments:

Page 2:
Line 6 – Ehn and Mundy (2013) is not really an appropriate citation for the statement.
Line 20 – 500 m

Line 29 – 1-cm... 1-m

Line 30 – Ice thickness or more appropriately, draft?

Line 32 – is this draft or thickness – if thickness, how do you account for ridge sails and snow cover? That is, a description of the methods needs to be included.

Line 2 – Fig. 1 c – From the figure, it is not clear how ice thickness sonar estimates agree with measurements.

Line 9 – A citation backing the statement on "more spatial precision" is needed as it is not clear why a statement would be true, provided the idea that you are seeking to cover a larger area with the AUV.

Line 16-18 – Why "therefore"? Does downsampled mean an average of 25 measurements? and what is the downsampled footprint being measured if the AUV is moving at 1.5 m s-1, with 75% of the measurement coming from a 4 m footprint at 25 Hz?

Line 19 – 10-nm

Line 22 – Depends on the algal pigment biomass in that algae can have a strong signal at 670 nm if enough biomass within the sea ice.

Line 23-28 – Why such a lengthy description here? I assume you took transmitted irradiance measurements over a depth profile to calculate water column attenuation coefficients for specific wavelengths. Or, did you use coefficients provided in Kirk (2011)? The more I read this paragraph, it was unclear.

Line 7 – Attached platelets? Were there unattached platelets? Was there ice algae on the platelets?

Line 18-19 – "passed through a nozzle of 60 ml syringe"? I have never heard of this method before. Is it common? I have a little concern that this method could act to burst cells? Typically, gentle back and forth inversion of the sample re-suspends the cells, which can then be subsampled. How were aliquots taken from the melted sample? Was the sample homogenized in terms of suspending the cells equally in the sample before subsampling?

Line 32 to Line 2 on Page 5 - ? It is not clear what a damaged core is and why one would not include the samples in the analysis. These details should be flushed out in the results when showing **all** the data to start.

Line 6-8 – Why even do the TriOS NDI calibration? Why not just compare the irradiance response of the two sensors together?

Line 17 – spelling - *Nitzschia frigida*

Fig. 2 – These are pretty incredible pictures. Visual inspection of the biomass shown in Fig. 2 looks to be very dense and growing up into the ice bottom beyond the saw in the picture. Is an average estimate of 26.7 mg m-2 realistic, or were these pictures from particularly high chl concentration sites?

Line 24 – R2 would be good to provide information on how much variability in chl concentration was described by the regression against the NDI values. It is included

in Fig. 3 caption, but why use an adjusted R2 value? The calculation for an adjusted R2 is to take into account the influence of more than one variable in a multiple regression, but only a linear simple regression is accomplished.
Line 25-31 – There appears to be a lot of discussion and citations in the results section. Is this appropriate for the journal?
Fig. 3 – Technically, the axes are plotted wrong. Chlorophyll should be the independent variable and NDI the dependent.

Line 23-24 – There appears to be a lot of interpretation and discussion in the results section – suggestion to split this out.

Line 4 – Why are there no comparisons with literature that already exists on ice algae patchiness? There were at least 5 studies in the 90s to early 2000s on this topic.
Line 30 – Is there a citation to support this statement? It makes sense, but has someone investigated this before?

Line 22 – ROVs do the same task, so it is not clear why this point is made. Actually, one can argue that the need to keep an AUV further from the ice bottom than an ROV would be a detriment to the technique due to the error associated with water column attenuation. The main benefit of an AUV is distance covered by a single deployment, which is not demonstrated in the current manuscript. Is the inclusion of the statement on line 22 valid given the results presented?